

# Combined use of karyotyping and copy number variation sequencing technology in prenatal diagnosis

Suhua Zhang[1], Yuexin Xu[1], Dan Lu[1], Dan Fu[1] and Yan Zhao[2]

[1] Department of Gynaecology and Obstetrics, Clinical Medical College of Yangzhou University, Northern Jiangsu People's Hospital, Yang Zhou, Jiangsu Province, China
[2] Medical Research Center, Clinical Medical College of Yangzhou University, Northern Jiangsu People's Hospital, Yang Zhou, Jiangsu Province, China

## ABSTRACT

**Background**. Karyotyping and genome copy number variation sequencing (CNV-seq) are two techniques frequently used in prenatal diagnosis. This study aimed to explore the diagnostic potential of using a combination of these two methods in order to provide a more accurate clinical basis for prenatal diagnosis.

**Methods**. We selected 822 pregnant women undergoing amniocentesis and separated them into six groups according to different risk indicators. Karyotyping and CNV-seq were performed simultaneously to compare the diagnostic performance of the two methods.

**Results**. Among the different amniocentesis indicators, abnormal fetal ultrasounds accounted for 39.29% of the total number of examinees and made up the largest group. The abnormal detection rate of non-invasive prenatal testing (NIPT) high risk was 37.93% and significantly higher than the other five groups ($P < 0.05$). The abnormal detection rate of mixed indicators was significantly higher than the history of the adverse reproductive outcomes group ($P = 0.0151$). The two methods combined found a total of 119 abnormal cases (14.48%). Karyotyping detected 57 cases (6.93%) of abnormal karyotypes, 30 numerical aberrations, and 27 structural aberrations. CNV-seq identified 99 cases (12.04%) with altered CNVs, 30 cases of chromosome aneuploidies, and 69 structural aberrations (28 pathogenic, eight that were likely pathogenic, and 33 microdeletion/duplication variants of uncertain significance (VUS)). Thirty-seven cases were found abnormal by both methods, 20 cases were detected abnormally by karyotyping (mainly mutual translocation and mostly balanced), and 62 cases of microdeletion/duplication were detected by CNV-seq. Steroid sulfatase gene (STS) deletion was identified at chromosome Xp22.31 in three cases. Postnatal follow-up confirmed that babies manifested skin abnormalities one week after birth. Six fetuses had Xp22.31 duplications ranging from 1.5 Kb to 1.7 Mb that were detected by CNV-seq. Follow-up showed that five babies presented no abnormalities during follow-up, except for one terminated pregnancy due to a history of adverse reproductive outcomes.

**Conclusion**. The combination of using CNV-seq and karyotyping significantly improved the detection rate of fetal pathogenic chromosomal abnormalities. CNV-seq is an effective complement to karyotyping and improves the accuracy of prenatal diagnosis.

Corresponding authors
Dan Fu, 15651057398@163.com
Yan Zhao, zhaoyan1982@foxmail.com

## INTRODUCTION

Karyotyping of amniotic fluid cells is still the most common technique used to identify chromosomal abnormalities and has been the gold standard in prenatal cytogenetic analysis. However, due to its long detection period and low detection resolution, it is unable to identify genomic copy number variations (CNVs) smaller than 10 Mb. CNVs are losses or gains of genomic segments and are a type of structural variation. CNVs are usually defined as genomic segments and present variable copy numbers that are 1 Kb or larger when compared with a reference genome (*Nevado et al., 2014*). A chromosomal microarray (CMA) is mainly used for the detection of genome-wide CNVs and plays a very significant role in the prenatal and postnatal samples for the detection of chromosomal aberrations (*Cheng et al., 2019*; *Wang et al., 2020*). However, due to its high cost, low throughput, and complex experimental procedures, the large-scale application of this technique in prenatal diagnosis is limited. Moreover, the limited coverage of the CMA probe presents the possibility that some pathogenic copy number variations (pCNVs) may not be detected (*Dong et al., 2016*; *Hayes et al., 2013*).

Based on the emergence of next-generation sequencing (NGS) technology, the detection of CNVs has a wider range, higher throughput, lower cost, shorter reporting period, and lower DNA sample requirements, making it more suitable for clinical applications (*Liang et al., 2014*; *Liu et al., 2015*; *Xie & Tammi, 2009*; *Zhang et al., 2021b*; *Zhu et al., 2016*). Therefore, we speculated that a simultaneous analysis and comparison of the results from karyotyping and genome copy number variation sequencing (CNV-seq) may be more effective in the diagnosis of chromosomal abnormalities and improve the accuracy of diagnosis. To explore this theory, we explore the possibility of using a combination of the two methods in the prenatal diagnosis of chromosomal abnormalities in 822 pregnant women who underwent traditional karyotype analysis and CNV-seq testing simultaneously.

## PATIENTS & METHODS

### Study patients and design

Between January 2017 and December 2021, 2,631 pregnant women with high-risk indicators underwent amniocentesis at the Department of Prenatal Diagnosis in Northern Jiangsu People's Hospital, China, and 822 of those pregnant women received karyotyping and CNV-seq simultaneously. This study was approved by the Medical Ethics Committee of Northern Jiangsu People's Hospital (No. J2014012, No. 2019095). All participants received genetic counseling and provided informed consent before testing, including maternal serum screening, ultrasound examination, and amniocentesis for detecting fetal chromosomal anomalies using karyotyping and CNV-seq.

When abnormalities were identified on the fetal chromosomes, the peripheral blood of the parents were collected for parental verification and prenatal diagnosis, to judge whether the abnormality was de novo or inherited. All pregnancy outcomes were recorded.

## Amniocentesis indicators

Study subjects were divided into six groups according to their indicators for amniocentesis, namely: advanced maternal age (AMA) ≥35 years and advanced paternal age ≥45 at the time of delivery, high risk determined by maternal serum screening (at least one positive item determined by mid-term serological screening; the risks for trisomy 21 and trisomy 18 were determined by measuring second-trimester serum markers and with scores of ≥1 in 270 and ≥1 in 350, respectively), abnormal fetal ultrasonography (including structural malformation and soft markers), history of adverse reproductive outcomes (including abortions, stillbirths, perinatal death, premature delivery, and congenital malformations), and high risk determined by NIPT (suggesting the existence of whole or partial chromosome duplication and deletion). The last group consisted of patients with mixed indicators, including the inheritable risk of a single gene disease, prior risk of an abnormal pregnancy outcome, chromosome abnormality carriers, intellectual disability of pregnant women, exposure to toxic substances during early pregnancy, or other diseases.

## Amniocentesis

Amniotic fluid samples were obtained from pregnant women through ultrasound-guided transabdominal puncture (*Izetbegovic & Mehmedbasic, 2013*), We collected 27 ml of amniotic fluid, discarded the first 2 ml to avoid contamination by maternal blood, and 20 ml of amniotic fluid was used for cell culture and karyotyping. The remaining 5 ml was directly used for CNV-seq without culture.

## Karyotyping

Amniotic fluid samples were cultured following the standard protocols. Chromosome preparations were G-banded using trypsin-Giemsa staining for karyotyping following a series of standard protocols including colchicine treatment, hypotonic treatment, fixation, and centrifugation. Chromosome karyotype map scanning and acquisition were done using an automatic metaphase chromosome analysis system (GSL-120: Leica Microsystems, Deerfield, IL, USA). Karyotypes were defined according to the international system of Human Cytogenetic Nomenclature (*McGowan-Jordan, Simons & Schmid, 2016*).

## CNV-seq

CNV-seq was conducted by a third-party laboratory, Berry Genomics Co. The process in brief was as follows: quality control of DNA in amniotic fluid cells using short tandem repeat (STR) markers in order to avoid contamination from maternal DNA. Genomic DNA was extracted using the QIAamp DNA mini kit (Qiagen, Valencia, CA, USA) following the manufacturer's instructions: 50 ng of amniocyte DNA was fragmented and DNA libraries were constructed by end repair, ligated with sequencing adaptors, and the modified fragments were amplified by polymerase chain reaction (PCR). DNA libraries

were subjected to massively parallel sequencing to produce approximately 5 million raw sequencing reads with genomic DNA sequences of 36 base pairs in length on the Nextseq 500 platform (Illumina, San Diego, CA, USA) (*Wang et al., 2018*). Sequencing results from each sample were mapped to the human genome reference hg19, and updated to the human genome reference GRCh38 (hg38) with DECIPHER database according to the ISCN 2020. The identified and mapped CNVs were interpreted according to publicly available databases, including the Database of Genomic Variants (DGV); Online Mendelian Inheritance in Man (OMIM); DECIPHER, University of California, Santa Cruz (UCSC); and PubMed. Their pathogenicity was assessed according to the guidelines outlined by the American College of Medical Genetics (ACMG) for the interpretation of copy number variants (*Kearney et al., 2011*; *Riggs et al., 2020*). CNVs were interpreted and divided into five categories: pathogenic, likely pathogenic, variants of uncertain significance (VUS), likely benign, and benign. To facilitate clinical interpretation, we only analyzed the first three types of CNVs in this study.

## Statistical analysis

To analyze clinical data, SPSS software version 24.0 (IBM Corp., Armonk, NY, USA) was used for statistics. Comparison of categorical data between groups was analyzed using Chi-square test. $P < 0.05$ was considered statistically significant.

# RESULTS

## Characteristics of subjects

The ages of the pregnant women ranged from 15 to 48 years, and the median age was $30.7 \pm 5.7$ years. There were 595 women aged <35 and 227 women aged $\geq$35 at the expected date of childbirth, with a gestational age of 15 to 31 weeks. There were 2 husbands aged $\geq$45.

Across the six groups with different risk indicators of amniocentesis detected by the two methods, abnormal fetal ultrasonography accounted for 39.29% of the total number of subjects. Advanced maternal age, maternal serum screening high-risk, history of adverse reproductive outcomes, NIPT high-risk, and mixed indicators accounted for 27.86%, 17.64%, 16.42%, 7.06%, and 8.76% of total subjects, respectively. The highest abnormal detection rate was in the NIPT high-risk group (37.93%). The abnormal detection rates of mixed indicators, maternal serum screening high-risk, abnormal fetal ultrasonography, advanced maternal age, and history of adverse reproductive outcomes were 20.83%, 15.17%, 13.93%, 11.79% and 8.89%, respectively. The statistical results showed that the abnormal detection rate of the NIPT high-risk group was significantly higher than all of the other groups ($P < 0.05$), and the abnormal detection rate of mixed indicators was significantly higher than the history of the adverse reproductive outcomes group ($P < 0.05$). We divided the 822 women into groups according to whether they had a single indicator or $\geq$2 indicators for prenatal diagnosis. The abnormal detection rate of $\geq$2 indicators was higher than that of the single indicators, and there was no significant difference between the two groups. The results are shown in Table 1.

**Table 1  Proportion of amniocentesis indicators and abnormal detection rates in each group by karyotyping and CNV-seq.**

| Indicators for prenatal diagnosis | Constituent rate (No. of cases/total cases) | Abnormal detection rate |
|---|---|---|
| AMA | 27.86% (229/822) | 11.79% (27/229) |
| Maternal serum screening high-risk | 17.64% (145/822) | 15.17% (22/145) |
| Abnormal fetal ultrasonography | 39.29% (323/822) | 13.93% (45/323) |
| History of adverse reproductive outcomes | 16.42% (135/822) | 8.89% (12/135) |
| NIPT high-risk | 7.06% (58/822) | 37.93% (22/58)[*] |
| Mixed indicators | 8.76% (72/822) | 20.83% (15/72)[**] |
| Single indicator | 83.45% (686/822) | 13.85% (95/686) |
| ≥2 indicators | 16.55% (136/822) | 17.65% (24/136) |
| Total | 100% (822/822) | 14.48% (119/822) |

Notes.

AMA, advanced maternal age; NIPT, non-invasive prenatal testing.

[*]NIPT vs AMA, $P < 0.0001$; NIPT vs maternal serum screening high-risk, $P = 0.0004$; NIPT vs abnormal fetal ultrasonography, $P < 0.0001$; NIPT vs history of adverse reproductive outcomes, $P < 0.0001$; NIPT vs mixed indicators, $P = 0.0317$.

[**]Mixed indicators vs history of adverse reproductive outcomes, $P = 0.0151$.

## Comparison of karyotyping and CNV-seq results

In this study, a total of 822 pregnant women underwent the standard karyotyping test and CNV-seq test simultaneously. There were 57 patients with abnormal karyotypes, 30 with numerical aberrations and 27 with structural aberrations (made up of 12 cases of balanced translocation, six cases of unbalanced translocation, five cases of inversion, one case of marker chromosome, one case of mosaic, and two cases of uncertain chromosome deletion/duplication). CNV-seq identified 99 cases with altered CNVs, accounting for 12.04% of the total. Among them, 30 cases had chromosome aneuploidies, 69 had structural aberrations (made up of 28 cases of pathogenic microdeletion/duplication, eight cases of likely pathogenic microdeletion/duplication, and 33 cases of VUS microdeletion/duplication).

The two methods combined found a total of 119 abnormal cases, which was 14.48% of the total. Thirty-seven cases were determined abnormal by both test methods (Table S1), and 30 cases were confirmed to have whole chromosome aneuploidy, and consisted of 16 cases of trisomy 21 (53.33%), four cases of trisomy 18 (13.33%), one case of trisomy 13 (3.33%), and nine cases of sex chromosome aneuploidies (SCAs) (30.0%, including three mosaics). The rest of the seven cases were confirmed to have pathogenic deletion/duplication. Except for case 37, the results of the karyotyping and CNV-seq were consistent. There were 63 cases of microdeletion/duplication detected only by CNV-seq. The details of the 22 pathogenic (including one pathogenic CNV case 37 in Table S1) and 8 likely pathogenic microdeletion/duplication cases are shown in Table S2. The details of the 33 VUS microdeletion/duplication cases are shown in Table S3. There were 20 cases with abnormal chromosome karyotypes that were not detected by CNV-seq (Table S4), including 11 cases of balanced translocation, five cases of inversion, one case of marker chromosome, one case of mosaic, and two cases of uncertain chromosome deletion/duplication. These clinical data suggested that the combined use of karyotyping and CNV-seq could improve the abnormal detection rate and the accuracy of prenatal diagnosis.

### Results of two special cases

In this study, the chromosomes of cases 22 and 39 were special. The prenatal diagnosis indication of case 22 was NT 3.8 mm, and the karyotype was 45, XN, rob (14; 21) (q10; q10). The translocation of chromosome 21 is larger than that of normal chromosome 21, and it is difficult to judge whether it is abnormal karyotype and the source of the abnormal fragment. Case 22 was confirmed as seq[GRCh38]dup(21)(q21.1q22.3)(19047682-46680088) × 3, which showed 27.68 Mb duplication in the 21q21.1q22.3 region and was consistent with trisomy 21. The prenatal diagnosis indicator of case 39 was maternal serum screening high risk (1/61), and the karyotype was 46, XN. However, case 39 was confirmed as seq[GRCh38]del(5)(p15.33p15.1) (20001-17939891) × 1, and seq[GRCh38]dup(7)(q34q36.3) (141680201-159335973) × 3, which showed 17.92 Mb deletion in the 5p15.33p15.1 region, and 17.76 Mb duplication in the 7q34q36.3 region. The results of karyotypes and CNV-seq of case 22 and 39 are shown in Figs. 1 and 2, respectively. Therefore, we concluded that CNV-seq could be used as an effective complement to karyotyping.

### Follow-up

Follow-up was conducted for the pregnancy outcomes of the 822 women. For pregnant women who underwent amniocentesis from 2017 to 2020, the babies completed the 1-year follow-up. For pregnant women who underwent amniocentesis in 2021, their babies completed from three months to six months followed-up. Of the 119 cases with abnormal results, 51 cases had terminated a pregnancy after informed consent. Most of them were induced by chromosome aneuploidy and pathogenic deletion/duplication. Cases 56–58 showed that these babies with the X-linked ichthyosis (XLI) gene didn't manifest skin abnormalities at birth. However, widely-distributed white scales present on the abdomen were aggravated in dry air one week following birth. Notably, fetuses of cases 91–96 showed 1.5 Kb to 1.7 Mb duplications in the Xp22.31 region, the clinical significance of duplications was not clear. Follow-up results showed that one pregnant woman chose to terminate her pregnancy due to an adverse pregnancy history; five babies were born at full-term delivery and four of them presented with nonphenotypes from birth to one year, one baby showed nonphenotypes from birth to six months. Follow up with case 48 found special facial appearance at birth, neonatal hypotonia, and growth retardation, which were consistent with 1p36 microdeletion. Case 67 had asphyxia at birth. Case 68's ultrasound examination showed that fetal hydronephrosis increased to 22 mm, and the fetal outcome was a stillbirth. One case was not able to be followed-up.

In the remaining cases with normal results, one case presented fetal death in utero during the third trimester of pregnancy, and in another case at 34 weeks of gestation, the fetus died in the womb. The remaining babies presented with nonphenotypes during follow-up.

## DISCUSSION

Today, abnormality of fetal chromosome number or structure is the main cause of fetal malformation, abortion, stillbirth, and neonatal death. Traditional cytogenetic analysis is quite limited when looking for submicroscopic chromosomal changes, namely CNVs.

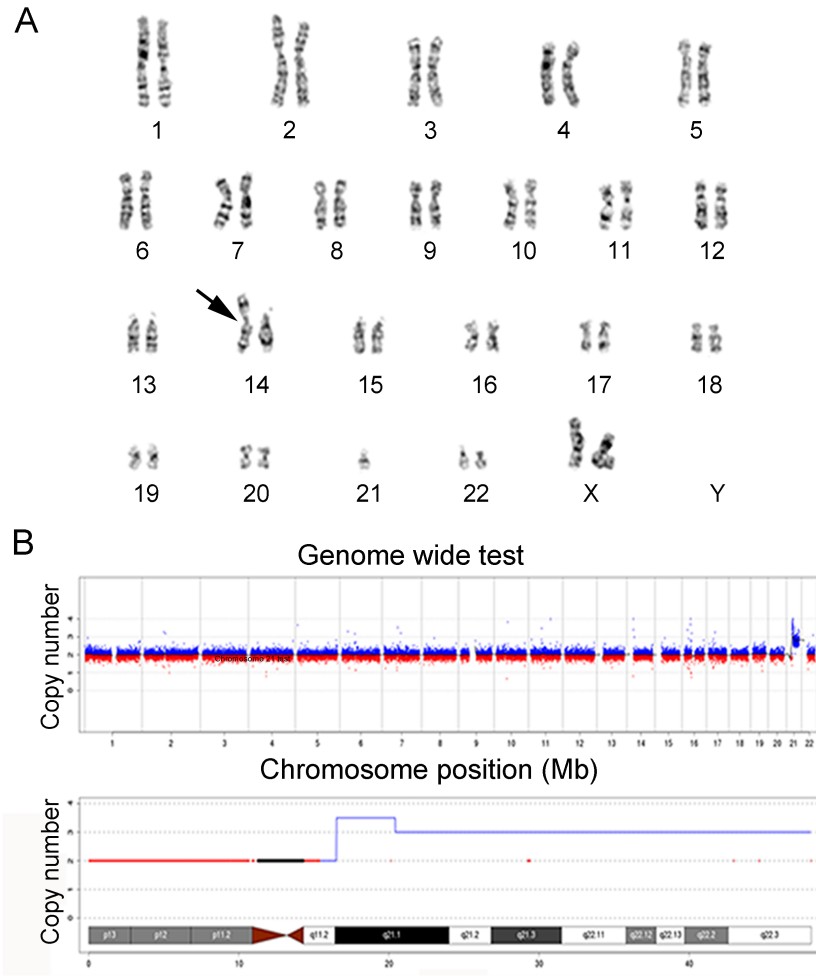

**Figure 1** **Karyotyping and CNV-seq results of case 22.** (A) Karyotype of case 22 with the abnormal chromosome indicated by arrow. The karyotype was 45, XN, rob (14; 21) (q10; q10). (B) CNV-seq was seq [GRCh38] dup (21) (q21.1q22.3), and q21.1-q22.3 repeated 27.68 Mb region, which is the key region of Down's syndrome.

So far, over 300 kinds of chromosome microdeletion/ duplication syndromes caused by pCNVs have been identified, with an incidence rate of 1/600 (*Goldenberg, 2018*; *Nevado et al., 2014*), accounting for half of the birth defects caused by chromosomal aberrations (*Evans, Wapner & Berkowitz, 2016*). Previous studies have shown that 6–7% of fetuses with normal karyotypes but abnormal structure indicated by ultrasound have definite or possible pathogenic CNVs (*Callaway et al., 2013*; *Hillman et al., 2013*; *Wapner et al., 2012*). Therefore, a combination of karyotyping and CNV-seq has gradually been used in prenatal diagnosis.

In this study, 822 women underwent traditional karyotype analysis and CNV-seq test. The highest constituent ratio of amniocentesis indicators was in abnormal fetal ultrasound (39.29%), those patients with abnormal ultrasound were more inclined to choose the combined detection of the two technologies (*Wang et al., 2018*; *Zhang et al., 2021a*; *Zhao &*

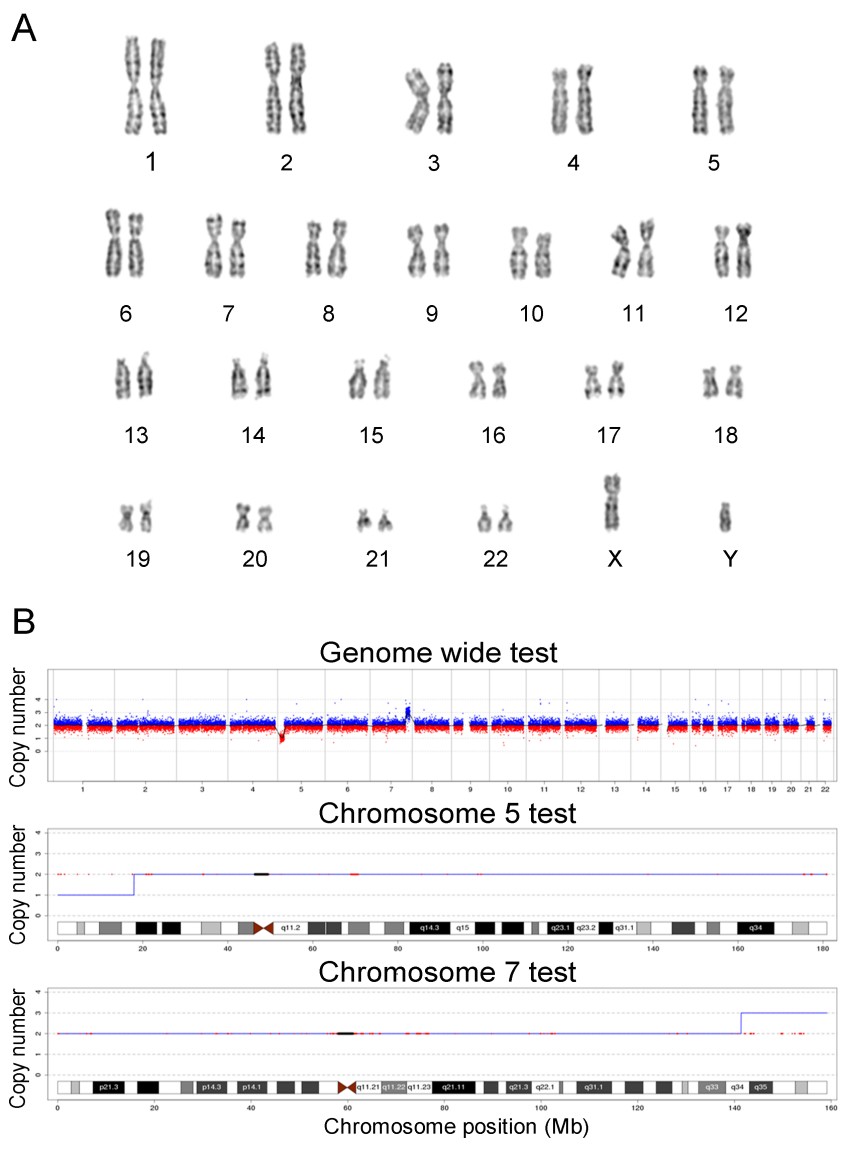

**Figure 2** **Karyotyping and CNV-seq results of case 39.** (A) The karyotype was 46, XN. (B) CNV-seq was seq [GRCh38] del (5) (p15.33p15.1), seq [GRCh38] dup (7) (q34q36.3), the 17.92 Mb region was deleted at p15.33-p15.1 on chromosome 5, located in the critical region of the Cri-du-chat Syndrome (5p deletion), and the 17.76 Mb region was duplicated at q34-q36.3 of chromosome 7.

*Fu, 2019*). Thus, abnormal fetal ultrasonography is a major indicator for prenatal molecular diagnosis. The abnormal detection rate was 13.93% in the group with prenatal ultrasound abnormalities, which was similar to the research results of *Wang et al. (2018)*.

We found that the abnormal detection rate of the NIPT high-risk group was significantly higher than all of the other groups, and the abnormal detection rate of mixed indicators was significantly higher than the history of the adverse reproductive outcomes group. NIPT is highly accurate and has been effectively and widely used as a prenatal non-invasive screening method (*Zhang et al., 2015*). In addition to trisomy 21, 18, and 13 routine screening,

NIPT has a certain detection effect on sex chromosome aneuploidy and fetal pathogenic microdeletion/duplication, and can provide clinical application value for subsequent prenatal diagnosis. However, since NIPT is also a sequencing technology platform, it has the same defects as CNV-seq technology and cannot detect fetal polymorphism, balanced translocations, polyploids, and other fetal structural abnormalities. The chromosome abnormal detection rate of a fetus in the mixed group of multiple indicators was higher, this was because some people in the group were carriers of chromosome balanced translocation.

In the present study, the abnormal detection rate of combination of two or more indicators group was not significantly higher than that of single indicator group. Previous study found that multiple prenatal diagnosis indicators could decrease the sensitivity but increase the specificity to predict fetal pathogenic CNV (*Zhang et al., 2021b*). We acknowledged that there were some limitations of this study including relatively small sample size and its retrospective nature, which may predispose the study to selection bias and issues with missing data. Therefore, possible prenatal predictive efficiencies of combined different indicators for pathogenic chromosomal abnormalities required additional investigation.

In general, these results and Chinese expert consensus on the application of low-depth whole genome sequencing in prenatal diagnosis (*Clinical Genetics Group of Medical Genetics Branch Chinese Medical et al., 2019*) suggest that karyotyping and CNV-seq could be recommended as first-line prenatal diagnosis methods for pregnant women with the six high-risk indicators.

Combination of CNV-seq and karyotyping significantly could improve the detection rate of fetal pathogenic chromosomal abnormalities. Two combined detection methods found a total of 119 abnormal cases, which made up 14.48% of the total. There were 57 cases of chromosomal abnormalities that were detected by karyotyping, accounting for 6.93% of the total subjects. There were 99 women who were confirmed to have chromosomal abnormalities (pathogenic, likely pathogenic, and VUS) that were detected by CNV-seq, accounting for 12.04% of the total subjects. Among these patients, the abnormal detection rate for the pathogenic and likely pathogenic variants was 8.03% (66/822). Compared with karyotyping, the abnormal detection rate of pathogenic/likely pathogenic CNV-seq was increased by 1.10%, which was similar to the results of other studies (*Wang et al., 2019*; *Wang et al., 2018*).

CNV-seq could accurately locate the abnormal fragments of karyotypes and provide more accurate genetic information during prenatal diagnosis and clinical genetic counseling. Our research results showed that CNV-seq could detect all chromosomal aneuploidy abnormalities, such as trisomy 21, trisomy 18, and sex chromosome abnormalities. The results of CNV-seq and karyotyping were consistent. Notably, case 22 had Robertsonian translocation between chromosomes 14 and 21, which is the most common Rb translocation found in humans. CNV-seq analysis showed a case of 21q21.1-q22.3 duplication (27.68 Mb), which is the key region in Down's syndrome. In case 29, it was impossible to analyze Y chromosomes or small supernumerary marker chromosomes (sSMC) using karyotype analysis. However, CNV-seq suggested that the Y chromosome was amphiploid, thus the clinical diagnosis was 47, XYY. In case 32, there was a balanced

translocation between chromosome 4 and chromosome 16 in the father of the fetus. Chromosome balanced translocation carriers are prone to produce unbalanced gametes (*Morin et al., 2017*; *Wang et al., 2016*), such as in this case of 16q21-q24.3 duplication (27.18 Mb) and 4q35.1-q35.2 deletion (6.52 Mb). The clinical phenotype obtained from the database query was consistent with that of fetal congenital heart disease (double outlet right ventricle, ventricular defect) indicated by B-ultrasound, which suggested that these were clinically relevant CNVs. NIPT in case 33 suggested that other chromosomal abnormalities might occur in the fetus. CNV-seq showed 8q22.1q24.3 duplication (48.5 Mb) and 12q24.33 deletion (1.1 Mb). The duplication region of 8q22.1q24.3 belongs to trisomy 8 (including the key segment of 8q22-q24). The clinical features include short stature, special facial features, cryptorchidism, hypertrichosis, congenital heart disease, intellectual disability, and frequent seizures. The CNVs were also clinically relevant.

CNV-seq can not only accurately locate the source of abnormal chromosomal fragments, but also find chromosomal microdeletions and microduplications that cannot be found by karyotyping. In this study, we found 22 cases of pathogenic CNVs, eight cases of likely pathogenic CNVs, and 33 cases of VUS that could not be detected by karyotyping. Among the 22 cases of pathogenic CNVs, cases 40 and 48 were the results of two pregnancies from the same woman, and the fetus was retained after informed consent of the second pregnancy. The clinical manifestations of this pregnant woman were mainly special facial features, skeletal dysplasia, and intellectual disability. The genotype of the mother was seq[GRCh38]del(1)(p36.33p36.32) NC_000001.11:g.884621_2823435del. The two fetal pathogenic CNVs inherited from the mother of the fetus could cause 1p36 microdeletion syndrome. It is worth mentioning that case 39 showed 17.92 Mb deletion in the 5p15.33p15.1 region (containing Cri du chat syndrome key genes), and 17.76 Mb duplication in the 7q34q36.3 region. Since karyotyping results do not indicate cytogenetic abnormalities, a confirmatory test such as FISH needs to be performed following CNV-seq. However, this pregnant woman refused to undergo FISH testing and chose to terminate her pregnancy voluntarily after genetic counseling.

Why did karyotype analysis not detect the chromosome abnormalities larger than 10 Mb? Due to the morphologic similarities between 5p15.33p15.1 and 7q34q36.3, it was difficult for karyotype analysis to accurately distinguish between subtle structural variations. We speculated that the short arm end of chromosome 5 was actually 7q34q36.3 translocation, and the two chromosomes 7 were normal.

Additionally, we found that the proportion of VUS microdeletion/duplication was 4.01% (33/822), which was slightly higher the results of *Wang et al. (2018)*. VUS presents challenges to clinical genetic counseling. Clinical intervention should be combined with the clinical phenotype and penetrance of CNV. Parental DNA testing by CNV-seq can help further interpret the pathogenicity of the fetal CNVs and define parental origin, so that the information could be used by the clinician to help interpret these VUS results, and manage these aneuploid pregnancies. In addition, most of the investigated VUS were proven to be *de novo* (*Wang et al., 2018*). For the clinicians and patients, the discovery of a de novo VUS is problematic and follow-up after birth is recommended by the ACMG (*Richards*

*et al., 2015*). In this study, we explored the pathogenicity of microdeletion/duplication in chromosome Xp22.31.

Cases 56–58 were male fetuses with deletion of 1.68 Mb regions on chromosome Xp22.31, which contains the entire steroid sulfatase gene (*STS*). Mutations and partial or entire deletions of *STS* have been reported to cause XLI (*Zhang et al., 2020*). The main clinical manifestation of XLI is large areas of scales on the limbs, face, neck, trunk, and buttocks. These skin lesions persist and do not improve with age. XLI occurs almost only in male patients at birth or shortly after birth. Due to the influence of potential factors such as X chromosome abnormal fragment size, connection position, random inactivation, bias inactivation, and gene escape inactivation, female carriers of X chromosome abnormal fragments may have some clinical variability. In our study, two cases were inherited from the maternal side and one case had familial ichthyosis. After genetic counseling, three pregnant women chose to continue the pregnancy with informed consent.

The fetuses of cases 91-96 had Xp22.31 duplications ranging from 1.5 Kb to 1.7 Mb. This region covered four genes: *PUDP, STS, VCX,* and *PNPLA4*. The pathogenicity of Xp22.31 duplication seems to be controversial. Previous reports have shown that some individuals with duplication of this region had varied degrees of neurological impairment, including growth retardation, intellectual disability, autistic spectrum disorders, hypotonia, seizures, psychomotor retardation, and mild special face (*Faletra et al., 2012*; *Pavone et al., 2019*; *Polo-Antunez & Arroyo-Carrera, 2017*). Some studies showed that duplication of Xp22.31 is a risk factor for abnormal phenotypes or benign variants (*Liu et al., 2011*; *Zhuang et al., 2019*). In this study, we found that five babies with Xp22.31 duplication did not present with phenotypes during follow-up. We will continue to follow up on these cases to observe if they may show clinical phenotypes consistent with the disease-causing genes as they age. Therefore, the duplications of Xp22.31 with recurrent duplication may serve as VUS.

CNV-seq could not detect the balanced translocation and inversion of chromosomes, which can be detected by karyotyping, but CNV-seq can detect whether the balanced translocation is accompanied by chromosome microdeletion/ duplication. The CNV-seq results of the 20 cases (Table S4) were normal, confirming that CNV-seq could not detect the balanced translocation and inversion of chromosomes, and there was no increase or decrease of pathogenic genetic material during chromosome rearrangement (*Cohen et al., 2015*; *Zhao & Fu, 2019*). Balanced translocations and inversions of chromosomes are important causes of reproductive abnormalities. Couples in whom one partner has a balanced translocation or inversion may have an overall high miscarriage rate resulting from unbalanced gametes (*Kaser, 2018*). Due to the high possibility of abnormal gametes, the risk of recurrent abortion and birth of children with abnormal chromosomes also increased. It is suggested that the appropriate fertility program be selected according to the specific situation, such as prenatal diagnosis after natural pregnancy or the use of preimplantation genetic diagnosis (PGD) technology to select normal embryo transfer (*Liu et al., 2015*).

CNV-seq could verify pathogenicity of sSMC in prenatal diagnosis. sSMC, also known as marker chromosome or extra abnormal structure chromosome, refers to the redundant chromosome with morphology that can be identified, but its characteristics and source

cannot be identified by traditional karyotyping technology (*Mcgowan-Jordan, Hastings & Moore, 2020*). In order to accurately determine the clinical phenotype and survival of the fetus, it was necessary to detect the fetuses with sSMC by cytogenetic and molecular methods. Previous studies have reported that the detection rate of fetal sSMC in prenatal diagnoses was 0.8–1.51‰ (*Huang et al., 2012*; *Huang et al., 2019*). In this study, only one sSMC was found in 2631 cases of amniocentesis by karyotyping. The CNV-seq results showed that it did not contain known human disease-related pathogenic genes, which presumably did not increase the risk of sub-representational abnormalities, and the patient opted to continue the pregnancy following clinical counseling.

## CONCLUSION

In conclusion, karyotyping and CNV-seq have their own advantages and disadvantages in prenatal diagnosis. Using a combination of CNV-seq and karyotyping significantly improved the detection rate of fetal pathogenic chromosomal abnormalities. CNV-seq is an effective complement to karyotyping and improves the accuracy of prenatal diagnosis. NIPT is a recommended non-invasive prenatal screening method for fetal chromosomal abnormalities. It is believed that with the widespread application of CNV-seq, the pathogenicity of more VUS microdeletion/duplication will be explored, as will the development of clinical genetic counseling.

### Funding
This work was supported by the project on Maternal and Child Health Talents of Jiangsu Province, China (F201944). The funders had no role in study design, data collection and analysis, decision to publish, or preparation of the manuscript.

### Grant Disclosures
The following grant information was disclosed by the authors:
Maternal and Child Health Talents of Jiangsu Province, China: F201944.

### Competing Interests
The authors declare there are no competing interests.

### Author Contributions
- Suhua Zhang conceived and designed the experiments, performed the experiments, analyzed the data, prepared figures and/or tables, authored or reviewed drafts of the article, and approved the final draft.
- Yuexin Xu performed the experiments, prepared figures and/or tables, and approved the final draft.
- Dan Lu conceived and designed the experiments, prepared figures and/or tables, and approved the final draft.

- Dan Fu conceived and designed the experiments, performed the experiments, analyzed the data, authored or reviewed drafts of the article, and approved the final draft.
- Yan Zhao conceived and designed the experiments, analyzed the data, prepared figures and/or tables, authored or reviewed drafts of the article, and approved the final draft.

## Human Ethics

The following information was supplied relating to ethical approvals (*i.e.*, approving body and any reference numbers):

The study was approved by the Medical Ethics Committee of Subei People's Hospital (No. J2014012, No.2019095).

## Data Availability

The clinical raw data and results are available in the Supplementary File.

## Supplemental Information

Supplemental information for this article can be found online at http://dx.doi.org/10.7717/peerj.14400#supplemental-information.

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
