# Peer review of "Combined use of karyotyping and copy number variation sequencing technology in prenatal diagnosis"

_PeerJ, doi:10.7717/peerj.14400_

## Round 0.1 · original submission · Major Revisions

· Academic Editor

Major Revisions

Your manuscript was considered interesting by the reviewers however they identified a number of issues that need to be addressed. The reviewers would like more clarification on your methodology, such as in how you classified the samples, more details on sample collection and on where the CNV-seq analysis was performed (third party laboratory versus in house). One of the reviewers also suggested that you perform Rapid Aneuploidy Detection (RAD) testing before CNV-seq, since it can detect all the abnormalities you list in table 2, and that FISH should be performed to confirm the karyotyping results in table 2. The reviewers would also like you to deposit the data into a public repository. Lastly the reviewers suggested that you should classify Variants of Unknown Significance (VUSs) in a different category, and not with the abnormal variants, and that the English language needs to be improved and the formatting of the reference needs to be updated.

Please, submit a detailed rebuttal which shows where and how you have taken all comments and suggestions into consideration. If you do not agree with some of the reviewers’ comments or suggestions, please explain why. Your rebuttal will be critical in making a final decision on your manuscript. Please, note also that your revised version may enter a new round of review by the same or by different reviewers. Therefore, I cannot guarantee that your manuscript will eventually be accepted.

Reviewer 1 ·

Basic reporting

The authors reported the genetic diagnostic rates of karyotyping and copy-number variation sequencing on 556 fetuses undergoing invasive prenatal diagnosis. While 26% of the cases had abnormal results by both tests, the majority of the abnormalities were detected only by one platform due to technical limitations (CNV-seq does not detected balanced translocations, karyotype does not have enough resolution for submicroscopic CNVS). The authors conclude the combination of the two methods improves pregnancy management.

Comments:
Sufficient literature and background/context are provided.
The format of the reference list needs to be updated. There are multiple instances of incorrect citations in text and also mix ups between first and last names. The English of the manuscript could be improved.

Experimental design

The research question is defined, relevant, and meaningful. However, the study lacks novelty as the applications of karyotyping and CMA (equivalent to CNV-seq for CNV detection) have been well described and these methods have been utilized in prenatal diagnosis for around 10 years. (ACMG published guidelines for CMA in prenatal applications in 2013).

As it is written, it seems like amniocentesis was performed separately for karyotyping and CNV-seq. Please clarify also if CNV-seq was performed on cultured cells or direct.

Validity of the findings

The authors should consider classifying VUS as a separate category and not grouped with abnormal results.

Line 304: The conclusion states karyotype and CNV-seq each have their own pros and cons. However, the authors then suggested CNV-seq as a reliable and accurate alternative to karyotyping. This is contradictory and inconsistent with the conclusion in the abstract which suggested a combined approached.

Additional comments

Line 72: “women carrying the single pregnancy” should be “women carrying a single pregnancy”
Line 76: “amount them” should be “among them”
Line 80-82: of the indications for testing, positive for ultrasonographic markers (including structural malformation and soft index abnormality) and adverse pregnancy outcomes (including abortions, stillbirths, perinatal death, premature delivery, and congenital malformations) overlap in structural abnormalities. Do the authors mean the latter are demised/abortuses only?
By “Soft index abnormality”, do the authors mean “soft markers”?

Line 107: suggest revising heading “Next-generation sequencing (CNV-seq)” to “Copy-number variation sequencing (CNV-seq)”

Line 110: “sells” should be “cells”
Line 114: “end filling” should be “end repair”
Line 122-124: How is the pathogenicity of CNVs assessed according to ACMG guidelines for SEQUENCE variants? Would the authors be referring to the CNV interpretation guidelines by ACMG instead? Similar on line 273.
Line 186: “species” is not a unit for chromosome microdeletion and microduplication syndromes

Reviewer 2 ·

Basic reporting

The authors describe the results of the joint application of karyotype and CNV-seq for prenatal diagnosis in a single center. To our knowledge, many similar studies have been published in recent years.
The strength of this work lies in the clinical information completeness is fine. The weakness lies in the lack of coherence and logic of the language. In addition, there are doubts about the method of sample classification and the statistical results; The discussion and conclusion sections also need to be strengthened.

1. The English language should be improved to ensure that an international audience can clearly understand your text. I suggest you have a colleague who is proficient in English and familiar with the subject matter review your manuscript or contact a professional editing service.
2. Please delete the unnecessary information in the supplement table, such as column H in rows 356, 455, and 476.

Experimental design

3. Please clarify whether only 556 pregnant women received a prenatal diagnosis during this study or whether 556 of all pregnant women who received prenatal diagnosis received both karyotyping and CNV-seq.

4. Please make sure that the sequencing data is available on the publicly accessible repository, and a link to the repository has been included in the "Method" section of the manuscript.

5. For a patient who was willing to undergo prenatal diagnosis, please clarify if the amniotic fluid samples required for karyotyping and CNV-seq were taken at the same time? If so, please revise the description of the sample collection method in line 108.

6. Please describe in detail the rules for categorizing cases with multiple high-risk factors. e.g., the case with AMA and high-risk of NIPT (case Y20203, Y20260), which group should it be subjected to.

7. Karyotyping is a well-established routine laboratory technique for prenatal diagnostic. Please simplify this part in the Method section (line91-105). Please clarify whether the CNV-seq testings were performed in the authors' hospital or a third-party medical laboratory. If it is the latter, it should be clearly stated.

8. In lines 122-124, the cited article was not used for interpreting the pathogenicity of CNV but for minor nucleotide variations. Please correct the references (doi:10.1038/s41436-019-0686-8).


9. All the chromosomal abnormalities in Table 2 could be detected by a rapid aneuploidy detection testing such as QF-PCR or PNBobs, so why not performing a RAD testing before CNV-seq especially for case with high-risk of common aneuploidy detected by NIPT.

Validity of the findings

10. Figure 2: Since karyotype results did not indicate cytogenetic abnormalities, a confirmatory test such as FISH needs to be performed after CNV-seq and the result should be added to the fig.2.

11.In line 147, the author stated that there was no significant difference among other groups beside the Abnormal fetal ultrasound group and AMA group. In fact, there was also a statistical difference between the AMA and NIPT groups or the Mixed indications group as well, please validate and explain these results.

Additional comments

12. Please update the CNV-seq results in the Tables according to the ISCN 2020.

13. According to the existing guidelines (ACOG/SMFM,2016; CCMG/SOGC,2018), Prenatal whole-genomics CNV analysis is recommended for a patient with a fetus with one or more structural abnormalities identified on ultrasonographic examination. Please clarify whether the authors consider it necessary to recommend CNV-seq to all pregnant women who is undergoing invasive prenatal diagnosis? Are there any disadvantages and please discuss them.

---

## Round 0.2 · Major Revisions

· Academic Editor

Major Revisions

Thank you for addressing the reviewers' comments. However, the reviewers still had a number of concerns. As stated in the previous review, more detail and clarification on your methodology needs to be provided, specifically how you grouped your cases. Additionally, one of the reviewers had concerns about the total number of cases in table 1, whether they match up to 822 as you stated, or 958 (which they mentioned in their review), and the concordance between the numbers in table 1 and the supplemental table you provided.

Please, submit a detailed rebuttal which shows where and how you have taken all comments and suggestions into consideration. If you do not agree with some of the reviewers’ comments or suggestions, please explain why. Your rebuttal will be critical in making a final decision on your manuscript. Please, note also that your revised version may enter a new round of review by the same or by different reviewers. Therefore, I cannot guarantee that your manuscript will eventually be accepted.

Reviewer 1 ·

Basic reporting

Sufficient literature and background/context are provided.

Experimental design

The research question is defined and relevant.

Validity of the findings

The conclusions are linked to the original research question.

Additional comments

Line 130: “Sequencing results from each sample were mapped to the human reference genome GRCh38/hg19”
The GRCh38 is not equivalent to hg19, instead hg38. Please specify the genome build used in this study.

Lines 134-135: The authors have revised references to the correct ACMG guidelines however needs to change the sentence to read: “Their pathogenicity was assessed according to the guidelines outlined by the American College of Medical Genetics (ACMG) the interpretation of copy-number variants”
The authors need to revise this sentence to refer to the correct ACMG guideline for the interpretation of CNVs, not SNVs.

Minor comments:

I don’t think the word “cytogenetic” needs to be in front of karyotyping.
Line 74-77: I suggest to reword this sentence:
“To explore this theory, we explore the possibility of using a combination of the two methods in the prenatal diagnosis of chromosomal abnormalities in 822 pregnant women who underwent traditional karyotype analysis and CNV-seq testing simultaneously.”
Line 102: remove “also”

Lines 226: “changes in chromosome sub-microstructure,” should be “submicroscopic chromosomal changes”
There were some errors in supplementary table 1.
Row 493: “eq[hg19]del(X)(p22.33q28)(mos) chrX:g.1_155270560del"
Row 571: please use consistent CNV nomenclature

Reviewer 2 ·

Basic reporting

In this revised manuscript, the author increased the number of cases and corrected some mistakes. However, I am still not satisfied with two points: firstly, English writing is still confusing in many places. There are many non-technical terms in the manuscript and supplemental table, such as the term “Inter arm inversion” (paracentric inversion), “intestinal echo enhanced” (echogenic bowel), and so on. Secondly, the logic and readability of the article still need to be addressed.

Experimental design

The authors did not elaborate on the details of the methodology for case grouping and its basis, making it difficult to validate the authenticity of the data and objectivity of the results.

Validity of the findings

The authors claim that this study included 822 patients and these cases were divided into six categories based on different prenatal diagnosis indications, however, the total number of cases in these six groups was 958 in Table 1. Furthermore, the case number in the subgroups in Table 1 did not match the case number in the supplemental table. For example,the case number in the AMA group was 229 in Table 1, however, there were only 225 cases of AMA in the supplemental table.

Additional comments

1.Line 135, “interpretation of sequence variants”, the research contents of this work was CNV, not sequence variants, please correct this sentence.

2.Line 211-213, “Notably, cases 91-96 showed duplications in the Xp22.31 region, and these five pregnant women chose to continue their pregnancies. One pregnant woman chose to terminate her pregnancy due to an adverse pregnancy history”, This sentence is ambiguous, please fix this sentence.

3.The author stated that the indication for Case 48 was maternal chromosome abnormalities, please provide the genotype and phenotype of the mother.

4.Line 221-222 “The remaining babies presented with nonphenotypes at 3 months, 6 months, and 1 year following their birth”. The author stated that the most recent patients were recruited in December 2021, how can they obtain follow-up results of all the patients for 1 year?


5.Line 262-264 “There were 99 women who were confirmed to have chromosomal abnormalities (pathogenic, likely pathogenic, and VUS) that were only detected by CNV-seq”. Please correct this sentence, as the author stated in the result section, 30 cases with aneuploidies and 7 cases with structural chromosomal abnormalities were detected both by CNV-seq and karyotyping.


6.Line 291-292, “we found 22 cases of pathogenic CNVs, eight cases of likely pathogenic CNVs, and 33 cases of VUS that could not be detected by cytogenetic karyotyping”, the number of P/LP cases was inconsistent with Table 3. Please explain.

7.Line 311-312, ”If the same situation was found in normal parents, it can be predicted as benign changes”. We could not agree with this conclusion cause factors such as variable expressivity and incomplete penetrance may influence the interpretation of hereditary CNV.

8.Line 344-345, “These fetuses did not show abnormal clinical phenotypes and had fertility in adulthood”. This sentence is baffling. It is not clear whether this refers to cases in the literature or this study.

9.Please reorganize the references in lines 352-356.

10. For case 32, please discuss why there was ambiguity in the karyotype description of the fetus when the father had been already known to be a balanced translocation carrier.

11.Please refine the subheadings in the discussion section

12.Please revised the title of Table 3-5 to make it clearer and more accurate.

---

## Round 0.3 · Minor Revisions

· Academic Editor

Minor Revisions

Thank you for thoroughly addressing the reviewers' comments. However, the reviewers still had some minor concerns. Although you updated the manuscript using the hg19 genome build, the reporting of the CNV results still states seq[GTCCh38]. Additionally, even though you stated in your rebuttal that "cytogenetic karyotype" was replaced with "karyotype", there are still some instances in the revised manuscript where "cytogenetic karyotype" is used. Lastly, please change the format of the CNV results in the supplementary files you provided so that it is consistent with the text and data shown in the revised manuscript.

Please, submit a detailed rebuttal which shows where and how you have taken all comments and suggestions into consideration. If you do not agree with some of the reviewers’ comments or suggestions, please explain why. Your rebuttal will be critical in making a final decision on your manuscript. Please, note also that your revised version may enter a new round of review by the same or by different reviewers. Therefore, I cannot guarantee that your manuscript will eventually be accepted.

Reviewer 1 ·

Basic reporting

Sufficient literature and background/context are provided.

Experimental design

The research question is defined and relevant.

Validity of the findings

The conclusions are linked to the original research question.

Additional comments

The authors have now changed the manuscript to state the genome build used is hg19 on line 132 but the CNV results are still listed as seq[GRCh38] throughout the manuscript.

Make the format of the CNV results in the supplementary files consistent with the manuscript.

The words "cytogenetic karyotype" were not revised as per suggested in the supplementary files.

---

## Round 0.4 · accepted · Accept

· Academic Editor

Accept

Thank you for addressing the reviewer's and my comments and thus improving your manuscript.

Reviewer 1 ·

Basic reporting

Sufficient literature and background/context are provided.

Experimental design

The research question is defined and relevant.

Validity of the findings

The conclusions are linked to the original research question.